# Superconductivity in the antiperovskite Dirac-metal oxide $Sr_{3-x}SnO$

Mohamed Oudah[1], Atsutoshi Ikeda[1], Jan Niklas Hausmann[1,2], Shingo Yonezawa[1], Toshiyuki Fukumoto[3], Shingo Kobayashi[3,4], Masatoshi Sato[5] & Yoshiteru Maeno[1]

Investigations of perovskite oxides triggered by the discovery of high-temperature and unconventional superconductors have had crucial roles in stimulating and guiding the development of modern condensed-matter physics. Antiperovskite oxides are charge-inverted counterpart materials to perovskite oxides, with unusual negative ionic states of a constituent metal. No superconductivity was reported among the antiperovskite oxides so far. Here we present the first superconducting antiperovskite oxide $Sr_{3-x}SnO$ with the transition temperature of around 5 K. $Sr_3SnO$ possesses Dirac points in its electronic structure, and we propose from theoretical analysis a possibility of a topological odd-parity superconductivity analogous to the superfluid $^3$He-B in moderately hole-doped $Sr_{3-x}SnO$. We envision that this discovery of a new class of oxide superconductors will lead to a rapid progress in physics and chemistry of antiperovskite oxides consisting of unusual metallic anions.

[1] Department of Physics, Graduate School of Science, Kyoto University, Kyoto 606-8502, Japan. [2] Department of Chemistry, Faculty of Mathematics and Natural Sciences, Humboldt-Universität zu Berlin, Brook-Taylor-Strasse 2, Berlin 12489, Germany. [3] Department of Applied Physics, Graduate School of Engineering, Nagoya University, Nagoya 464-8603, Japan. [4] Institute for Advanced Research, Nagoya University, Nagoya 464-8601, Japan. [5] Yukawa Institute for Theoretical Physics, Kyoto University, Kyoto 606-8502, Japan. Correspondence and requests for materials should be addressed to M.O. (email: oudah@scphys.kyoto-u.ac.jp) or to S.Y. (email: yonezawa@scphys.kyoto-u.ac.jp) or to Y.M. (email: maeno@scphys.kyoto-u.ac.jp).

Oxides with perovskite-based structures have had one of the central roles in condensed-matter research for decades. In particular, the discoveries of high-temperature superconductivity in cuprates[1] and unconventional superconductivity in the ruthenate $Sr_2RuO_4$ (ref. 2) have driven the science community to deepen the concepts of strongly correlated electron systems substantially. These research efforts led to the discoveries of novel phenomena also in cubic perovskite oxides that are in the base of these materials. Examples are colossal magnetoresistance[3], multiferroicity[4] and super-conductivity with the transition temperature $T_c$ of up to 30 K, reported for $Ba_{0.6}K_{0.4}BiO_3$ (ref. 5). This history of perovskites tells us that the finding of a new class of oxide superconductors has a potential to initiate unexplored research fields.

Perovskite oxides have their counterparts, antiperovskite oxides $A_3BO$ (or $BOA_3$), in which the position of metal and oxygen ions are reversed. Antiperovskite oxides were first found accidentally in an attempt to produce $Sr_3Sn$, which turned out to be stable only with inclusion of oxygen, forming $Sr_3SnO$ (ref. 6). Upon this discovery, various $A_3BO$ with $A = Ca$, Sr, Ba and $B = Sn$, Pb, were synthesized and their structure identified. Antiperovskite oxides crystallize in cubic or pseudo-cubic structures with reverse occupancy of metal and oxygen relative to their perovskite counterparts, as illustrated in the inset of Fig. 1. Moreover, a recent study expanded antiperovskite oxides beyond the elements mentioned above[7]. In the case of $Sr_3SnO$, the Sr–Sr distance of 3.548 Å is close to the Sr–Sr distance in the similarly coordinated SrO, 3.650 Å (ref. 8), and is shorter than that in pure Sr, 4.296 Å (ref. 9). This comparison confirms that an oxidation state of $Sr^{2+}$ is realized in $Sr_3SnO$. With an assumption that O ions have $2-$ valency, we obtain a charge-balanced formula of $(Sr^{2+})_3(Sn^{4-})(O^{2-})$, indicating an unusual negative oxidation state of group-14 elements characteristic of antiperovskite oxides. The antiperovskite oxides were left relatively unexplored for some time, perhaps because of their instability in air. In particular, no superconductivity is reported among antiperovskite oxides. For a material related to antiperovskite oxides, the effect of interstitial oxygen on the superconducting properties of $La_3In$ ($T_c = 10$ K) has been reported[10]. The range of oxygen content was suggested to be limited much below unity. Moreover, $T_c$ remained identical to that of $La_3In$ and just a strong reduction of the diamagnetic volume fraction was reported by adding oxygen. Thus it was not demonstrated that the antiperovskite $La_3InO$ is a bulk superconductor. We comment here that there have been reports on superconductivity in antiperovskite carbides, nitrides and phosphides in recent years[11–14]; starting from the discovery in $MgCNi_3$ ($T_c = 8$ K)[11]. These compounds have conventional valence states in contrast to antiperovskite oxides, and the superconductivity is attributed to ordinary phonon-mediated pairing among transition-metal $d$ electrons.

Recent band calculations for a closely related antiperovskite oxide $Ca_3PbO$ reveal slightly gapped three-dimensional Dirac cones in the very vicinity of the Fermi level, owing to the energy-level inversion of the Ca-$3d$ and Pb-$6p$ bands near the $\Gamma$ point[15]. In particular, bands with the Dirac dispersion are dominant around the Fermi level, being ideal to investigate Dirac-electron properties. Also, it has been predicted that a class of topological crystalline insulators exists within antiperovskite oxides, and $Sr_3SnO$ lies on the border of the topological phase[16].

In this article, we present the evidence for bulk super-conductivity in polycrystalline $Sr_{3-x}SnO$ samples through observation of zero resistivity and Meissner diamagnetism. Superconducting samples are found to have dominant hole carriers with the carrier density of $1 \times 10^{27}\,m^{-3}$ at 300 K. We propose possible realization of a topological odd-parity superconducting state, which can have the condensation energy comparable to that of the ordinary $s$-wave superconductivity with the aid of the mixing of Sr-$4d$ and Sn-$5p$ orbitals (as outlined in Methods section 'Topological superconductivity').

## Results

**Sample preparation and characterization.** The powder X-ray diffraction of superconducting samples reveals cubic $Sr_{3-x}SnO$ to be dominant as presented in Fig. 1 but with some splitting in the peaks. The main phase has the lattice parameter $a = 5.1222$ Å and the minor phase $a = 5.1450$ Å. These values are compared with the reported value $a = 5.12$ Å for polycrystalline samples synthesized at 600–700 °C (ref. 4) and 5.1394 Å for single crystals synthesized at temperatures up to 1,100 °C (ref. 6). The phase splitting is most likely due to different Sr deficiencies in some part of the sample, as Sr partially evaporates during the reaction. In addition, a very small amount of Sn and other impurity phases containing Sr are present (marked with asterisks in Fig. 1). These

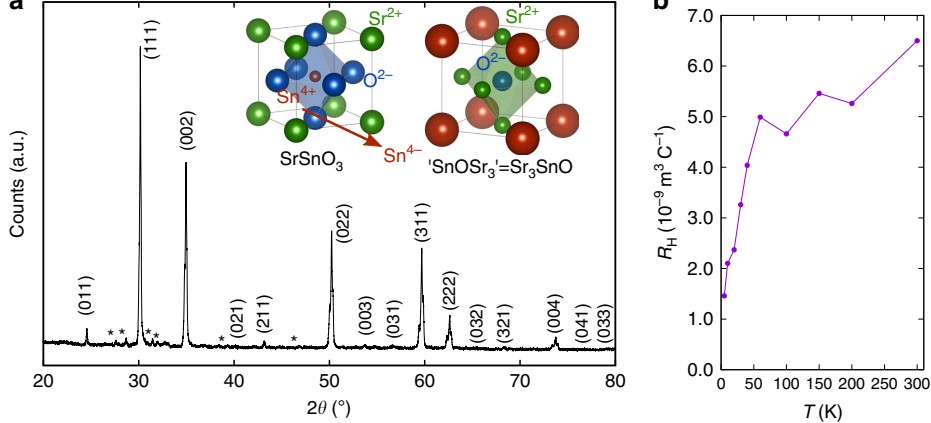

**Figure 1 | Structure and the sign of carriers in $Sr_{3-x}SnO$.** (**a**) Powder X-ray diffraction spectrum of a superconductive $Sr_{3-x}SnO$ batch-A sample. The spectrum was taken at room temperature on a lightly crushed sample with Kapton film and vacuum grease protecting it from air. Some impurity peaks can be seen marked with asterisks. The inset compares the perovskite $SrSnO_3$ and antiperovskite $Sr_3SnO$, emphasizing the change in oxidation states: $Sr^{2+}$ corresponds to $Sn^{4-}$, $Sn^{4+}$ to $O^{2-}$, and $O^{2-}$ to $Sr^{2+}$. (**b**) Hall coefficient measured as a function of temperatures showing hole-like carriers for a batch-A sample.

features in X-ray diffraction are common in other batches. All the samples showing superconductivity had Sr deficiency, either caused by significant Sr evaporation during synthesis or by Sr-deficient starting composition (see Methods section). As stoichiometric $Sr_3SnO$ does not show superconductivity, we conclude that a deficiency of strontium leads to superconductivity in this compound.

**Superconducting properties**. The resistivity of $Sr_{3-x}SnO$ exhibits a sharp drop with an onset of 4.9 K and zero resistivity below 4.5 K in zero field (Fig. 2a, batch C). Magnetic shielding of another sample from the same batch with an onset of 4.8 K is

observed in the imaginary and real parts of the alternating current (AC) magnetic susceptibility $\chi''_{AC}$ and $\chi'_{AC}$ (Fig. 2b,c). The transition in the direct current magnetization $M(T)$ obtained with zero-field-cooling processes is more pronounced than the curve for the field-cooling process, as typical for type-II super-conductivity with flux pinning. A sizable magnetic-flux expulsion in $M$ corresponding to the superconducting volume fraction of 32% at 2 K was observed for the same sample (Fig. 2d, Batch C). The $M(T)$ of another sample with the onset of 4.8 K exhibits a volume fraction of 62% at 2 K (Fig. 2d, Batch D). A similar data set for a sample from a different batch is also presented in Supplementary Fig. 1. These results assure bulk superconductivity in $Sr_{3-x}SnO$. The field dependence of $M$ reveals hysteretic behaviour again characteristic for a type-II superconductor (Supplementary Fig. 2). We note that the $\chi_{AC}$ curve obtained with adiabatic demagnetization cooling down to 0.15 K exhibits an additional transition at 0.8 K (Supplementary Fig. 1b,c). As the 0.8 K transition is reproducible in all superconducting batches that we investigated down to 0.15 K, it probably originates from another superconducting phase of $Sr_{3-x}SnO$ with different stoichiometry. The transition in the specific heat $C_P$ is not as pronounced due to the inevitably large contribution of the pho-non-specific heat compared with the electronic contribution with the small Sommerfeld coefficient $\gamma$. Nevertheless, a tiny anomaly below 5.1 K is observed after subtraction of the phonon con-tribution. The expected specific-heat jump is about 1% of the total specific heat and is on the order of uncertainty in the present measurements.

The superconducting phase diagram is shown in Fig. 3. Here, $T_c$ values were obtained from 10% and 50% resistivities shown in Fig. 2a. The upper critical field $H_{c2}$ for $T \to 0$ is estimated to be $\mu_0 H_{c2}(0) = 0.44$ T, using the Wertheimer–Helfand–Hohenberg relation $H_{c2}(0) = -0.72 T_c (dH_{c2}/dT)|_{T=T_c}$ (ref. 17). This value corresponds to the Ginzburg–Landau (GL) coherence length $\xi_{GL}(0)$ of 27 nm.

**Normal-state properties and electronic states**. The resistivity $R(T)$ shows metallic behaviour from room temperature with a relatively high residual resistivity ratio of $\sim 16$ and a small residual resistivity of $62 \mu\Omega$cm for a polycrystalline sample (Supplementary Fig. 3a). This contrasts with the semiconducting behaviour reported for $Sr_3SnO$ thin films[18,19]. The temperature dependence of the Hall coefficient $R_H$ (Fig. 1b) indicates that holes are the dominant carriers in the whole temperature range. The estimated carrier densities, if we assume a single hole band, are $4 \times 10^{27} m^{-3}$ at 5 K and $1 \times 10^{27} m^{-3}$ at 300 K. The

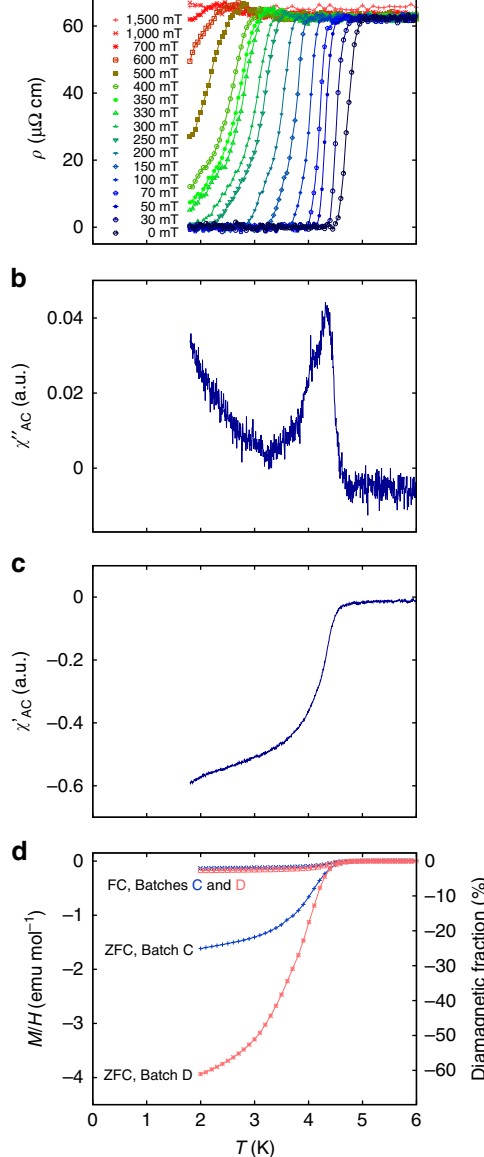

**Figure 2 | Superconducting transition of $Sr_{3-x}SnO$. (a)** Resistivity $\rho$ of a batch-C sample under zero and various magnetic fields. **(b,c)** Imaginary and real parts of AC susceptibility, $\chi''_{AC}$ and $\chi'_{AC}$, under zero magnetic fields for a batch-C sample. **(d)** Magnetization under zero field cooling (ZFC) and field cooling (FC) with an applied field of 0.5 mT of the same sample used in the $\chi_{AC}$ measurement (Batch C, blue crosses). Magnetization of a batch-D sample under ZFC and FC with an applied field of 1.0 mT is shown as well (red stars). The vertical scale on the right indicates the estimated diamagnetic volume fraction.

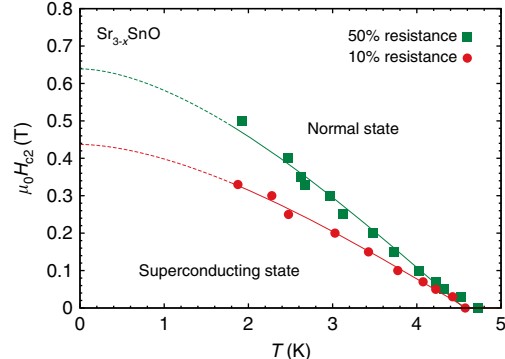

**Figure 3 | Field-temperature phase diagram of superconductivity in $Sr_{3-x}SnO$.** The upper critical field $H_{c2}$ is extracted from 10% and 50% resistivities for a batch-C sample. The curves are results of fitting with the Wertheimer–Helfand–Hohenberg relation[17].

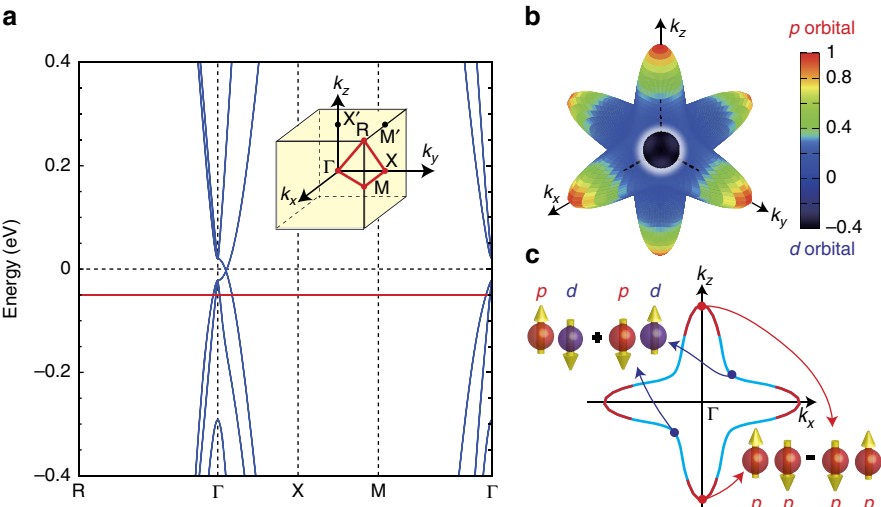

**Figure 4 | Orbital texture and possible Cooper pair symmetry of Sr$_{3-x}$SnO.** (**a**) Band structure of Sr$_3$SnO from tight-binding calculations with inverted orbital character and a Dirac point near the Γ point on each Γ–X line. Parameters are obtained by fitting to the first-principles calculation[16]. Inset shows the cubic Brillouin zone. (**b**) Orbital texture of the Fermi surface (FS) around the Γ point, reflecting the band inversion. The colour represents $\left(|\psi_p|^2 - |\psi_d|^2\right)/\left(|\psi_p|^2 + |\psi_d|^2\right)$, the degree of mixing of Sn-5p and Sr-4d orbital wavefunctions $\psi_p$ and $\psi_d$ at each k point on the FS. The red and black colours represent pure p and dominant d, respectively. Orbital mixing on the outer FS is strongest along the Γ–M direction, while the p orbital dominates in the Γ–X direction. (**c**) Possible Cooper pair symmetries. If superconducting symmetry is dictated by the pairing in the blue region on the FS, odd-parity spin-triplet pairing is favoured owing to orbital mixing. In case it is dictated by the red region, even-parity spin-singlet pairing is favoured.

magnetic field dependence of the Hall resistance is shown in Supplementary Fig. 3b, and the transverse magnetoresistance is presented in Supplementary Fig. 3c,d.

Such high-carrier density is consistent with heavy hole doping owing to Sr deficiency (Methods section). For stoichiometric Sr$_3$SnO, the electronic states near the Fermi level are influenced by the band inversion between Sr-4d and Sn-5p orbitals and the associated Dirac points near the Γ point[16], as shown in Fig. 4a. Upon hole doping, a Fermi surface (FS) originating from these Dirac points merges into one FS around the Γ point. This FS with a deformed-octahedral shape has an unusual orbital texture as shown in Fig. 4b. In addition, another hole pocket with orbital mixing centred at the Γ point appears inside (Supplementary Fig. 4). We note that a first-principles band calculation utilizing the Heyd–Scuseria–Ernzerhof screened Coulomb hybrid density functionals indicates yet another hole pocket around the R point[16]. In contrast, neither of our calculations based on the Heyd–Scuseria–Ernzerhof functionals nor the Perdew–Burke–Ernzerhof generalized gradient approximation reproduces the R-point pocket (Supplementary Fig. 5).

## Discussion
Besides being the first superconductor among the antiperovskite oxides, Sr$_{3-x}$SnO has the prospect of topological superconductivity. When Cooper pairs are formed in Sr$_{3-x}$SnO, their parity reflects the orbital texture of the underlying FS. Thus electrons on the FS portion with strong orbital mixing favour forming odd-parity and correspondingly spin-triplet pairs. This odd-parity state belongs to the same representation as that of the fully gapped superfluid $^3$He-B phase. As depicted in Fig. 4c, Cooper pairs can have either purely p or d–p mixed orbital character depending on the location on the outer and inner FSs. At present, we cannot deny the possibility that the hole FS around the R point, in the heavily hole-doped Sr$_{3-x}$SnO, is the main origin of the observed superconductivity. Even in that case, it is expected that pairing amplitude appears on the FS originating from the Dirac points and leads to unconventional

properties related to topological superconductivity[20,21]. Techniques used to observe Majorana zero modes on the surface of In$_{1-x}$Sn$_x$Te (ref. 22), a leading candidate for three-dimensional topological superconductors, may be adopted on Sr$_{3-x}$SnO to prove topological superconductivity.

In this article, we report evidence for bulk superconductivity in Sr$_{3-x}$SnO with an onset of about 5 K, marking the first superconductivity among antiperovskite oxides. Mirroring the rich variety of properties in perovskite oxides, the present work opens a door to superconductivity as well as other interesting phenomena in antiperovskite oxides with unusual metallic anions.

## Methods
### Sample synthesis
(1) Batches A, B and C. Bulk polycrystalline samples of Sr$_{3-x}$SnO, approximately 0.5 g in each batch, were synthesized by reaction of Sr chunk (Furuuchi, 99.9%) and SnO powder (Furuuchi, 99.9%) in an alumina crucible sealed inside a quartz tube under vacuum. Preparations of synthesis were performed inside an Ar-filled glove box. The sealed quartz tubes were heated to 800 °C over 3 h and kept at 800 °C for 3 h. Then the tubes were immediately quenched in water and were opened inside the glove box. The obtained samples were stored and prepared for measurements in the glove box. A crude estimation of the amount of evaporated strontium by weight in batches A, B, C indicates that it was about 18% of the starting strontium amount.
(2) Batch D. In further investigations on the synthesis, we observed that the evaporation of strontium can be suppressed to <1% if the reaction was carried out in glass tubes that were sealed under 0.3 bar of argon pressure at room temperature instead of vacuum. Batch D was synthesized in this way with Sr$_{2.46}$SnO as starting composition instead of Sr$_3$SnO.
(3) Batch E. An excess of 25% strontium was used and the glass tube was sealed under vacuum for synthesis of batch E. For this batch, a 100% yield of Sr$_3$SnO by weight with respect to the starting quantity of SnO was obtained. This batch E was considered an almost stoichiometric Sr$_3$SnO phase. A sample from batch E, with approximately stoichiometric composition, showed semiconducting resistivity behaviour down to low temperature, with an anomaly at 4 K, suggesting an inclusion of a small superconducting region. However, in the magnetic measurement, this sample did not show evidence of superconductivity down to 0.15 K. The observed semiconducting behaviour agrees with the result from Sr$_3$SnO thin-film reports[18,19] and supports our claim that the Fermi level is shifted down in the bulk superconducting samples.

**Powder X-Ray diffraction.** The powder X-ray diffraction measurements were performed on powder or lightly crushed samples with a diffractometer (Bruker AXS, D8 Advance) utilizing Cu-K$\alpha$ (1.54 Å) radiation selected by a Ni-mono-chromator. The sample was placed on a sample stage made with glass inside a glove box. Then the sample was covered with a 12.5-$\mu$m-thick polyimide film (DuPont, Kapton) attached to the sample stage with vacuum grease (Apiezon, N-grease) to prevent the samples from air contact during the measurements. We confirmed that the sample degradation is negligible with this setup within typical measurement time of 200 min. A broad X-ray diffraction peak originating from the Kapton film was observed only below $2\theta = 20°$. Structure refinement was performed using TOPAS package (Bruker AXS, Version 4-2).

**Characterization of superconductivity.** Transport, AC magnetic susceptibility and heat capacity measurements were carried out in a commercial apparatus (Quantum Design, PPMS). For the resistivity and Hall coefficient measurements, a five-probe method was typically employed from 1.8 to 300 K with fields up to 4.5 T. On a sample with a typical size of $1.8 \times 1.8 \times 0.6$ mm$^3$, 50-$\mu$m-diameter gold wires were attached using silver epoxy (EPOXY TECHNOLOGY, H20E) inside a He-filled glove box. The samples were protected from exposure to air with vacuum grease (Apiezon, N-grease) before taken out from the glove box. The AC susceptibility was measured using a small susceptometer compatible with the PPMS and its adiabatic demagnetization refrigerator option[23]. The specific heat of a piece of a sample covered with the grease was measured using a relaxation-time method using the PPMS. Additional small amount of grease, necessary to achieve good thermal contact to the sample, was applied to the sample stage during the addenda measurement as well. The addenda heat capacity was measured with and without the magnetic field to perform proper subtraction to obtain sample heat capacity.

The direct current magnetization $M$ was measured using a commercial SQUID magnetometer (Quantum Design, MPMS). Samples were sealed inside plastic capsules under Ar environment. Degaussing of PPMS and MPMS prior to measurements ensured the accuracy of the applied field values. The remnant field inside the MPMS was occasionally measured using a Pb (99.9999%) reference sample, and it was found to be $\leq 0.1$ mT after degaussing.

**Sample decomposition.** After exposed to air overnight, the sample decomposes into $Sr(OH)_2$ and Sn metal as confirmed by X-ray diffraction. Such sample exhibits a superconducting transition with $T_c = 3.7$ K as expected for pure Sn.

**Band-structure calculation.** The tight-binding model simplifies the calculation of orbital texture and enables us to study possible pairing. The band structure and the FS of $Sr_3SnO$ were calculated from tight-binding model, with the model parameters chosen to fit the band spectrum near the $\Gamma$ point of the first-principles calculation performed by Hsieh et al.[16] The tight-binding model was constructed in a manner similar to that for $Ca_3PbO$ (ref. 15); it consists of 12 orbitals, 6 of which originate from the Sr-4$d$ orbitals and the rest comes from the Sn-5$p$ orbitals.

**Topological superconductivity.** From the orbital texture of the FS around the $\Gamma$ point, a Cooper pair of $Sr_{3-x}SnO$ may form between electrons in different orbitals, when the effective pairing interaction is dominated by an attractive inter-orbital interaction. The resultant inter-orbital Cooper pair realizes an odd-parity superconducting state as Sn-5$p$ and Sr-4$d$ orbitals have opposite parity under inversion. Detailed theoretical analysis using the tight-binding model shows that the odd-parity pairing state is spin-triplet. The pairing symmetry is consistent with the $\Gamma_1^-$ representation in the cubic point group, which is the same representation as the fully gapped $^3$He-B phase. Thus the odd-parity state is also fully gapped, although it has dips in the gap along the $\Gamma$–X direction owing to suppression of orbital mixing in this direction. From the transformation law under the mirror reflection and four-fold rotation, the mirror Chern numbers on the $k_i = 0$ planes ($i = x, y, z$) can be evaluated. It is found that the mirror Chern numbers are 2 (mod. 4), implying the topological crystalline superconductivity of $Sr_{3-x}SnO$.

Furthermore, if the hole FS exists around the R-point, as the first-principles calculation suggested[16], the FS criterion of topological superconductivity[24–26] indicates that the odd-parity superconducting state is topological superconductivity. On the other hand, if the effective pairing interaction is dominated by an attractive intra-orbital interaction, an $s$-wave pairing symmetry may be realized. We note that a similar competition between different pairing states occurs in $Cd_3As_2$, where an odd-parity solution of the gap equation indeed has the critical temperature comparable to that of the ordinary $s$-wave pairing state[27,28].

**Data availability.** The data that support the findings of this study are available from the corresponding authors upon reasonable request.

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

## Acknowledgements

We thank discussions with J. Georg Bednorz, Yukio Tanaka and Masaaki Araidai. We also thank technical support from M.P. Jimenez-Segura, M.S. Anwar, C. Sow, T. Watashige, Y. Kasahara, S. Kasahara, Y. Matsuda and M. Maesato and Supercomputer Center, the Institute for Solid State Physics, the University of Tokyo for the use of the facilities. This work was supported by the JSPS KAKENHI Nos. JP15H05851, JP15H05852, JP15H05853 and JP15H05855 (Topological Materials Science), as well as by Izumi Science and Technology Foundation (Grant No. H28-J-146).

## Author contributions

Y.M. designed the project; S.Y. supervised most aspects of the experiments; M.O., A.I., J.N.H., S.Y. and Y.M. participated in sample preparation, measurements and data analysis; In particular, the synthesis of batches A, B and C and magnetic and thermal measurements were mainly performed by M.O., while the transport measurements by A.I. and the synthesis of batches D and E was mainly performed by J.N.H.; T.F., S.K. and M.S. performed band calculations and theoretical analysis of the pairing symmetry;

M.O., A.I., J.N.H., S.Y., M.S. and Y.M. wrote the manuscript with contributions from T.F. and S.K.

## Additional information

**Competing financial interests:** The authors declare no competing financial interests.

