## [Peer Review File · Nature Communications]

Reviewers' comments:

Reviewer #3 (Remarks to the Author):

The paper reports the synthesis of a possible new superconductor Sr₃SnO which has an anti-perovskite structure. The authors claim that it might be a superconductor with the Dirac metal feature in the normal state. A simple assessment on the data will find that the conclusions are not supported by the data and analysis. This does not allow me to give a positive judgement. Let me address the point below:

1. About the superconductivity. The XRD data seems to tell that the material Sr₃SnO might be successfully synthesized. The feeling from the XRD data would suggest a high percentage of the material of Sr₃SnO. While the DC ZFC magnetization would tell that the total superconducting magnetic screening volume at 2K is about 15%. In addition, a clear kink can be seen at about 3.7K which is most possibly due to the superconductivity of Sn impurity. The existence of Sn impurity has been confessed by the authors. Taking this diamagnetic signal away, one sees that the so-called diamagnetic signal from the possible superconductivity at 5K is really weak, which cannot be coherently explained as due to the major phase of Sr₃SnO.

2. To prove the material is a Dirac metal, one needs elegant band structure calculation and spectroscopy experiment. Only the transport data cannot give a definite support to the conclusion.

Based on these two key critics, I do not recommend the publication of this paper.

Reviewer #4 (Remarks to Authors):

Re: Superconductivity in the antiperovskite Dirac-metal oxide Sr₃SnO by Oudah et al.

Dear Editor,

The manuscript reported the discovery of superconductivity with $T_c \approx 5$ K in an antiperovskite compound Sr₃SnO via measurements of resistivity and AC/DC magnetic susceptibility. Despite of the absence of obvious specific heat jump at T_c , the occurrence of bulk superconductivity has been substantiated by the observations of zero resistivity and the Meissner diamagnetism in this compound. This makes Sr₃SnO the first case of superconductor among the family of antiperovskite oxides. In addition, the authors have proposed a possible topological superconductivity based on the first-principles calculations and theoretical analysis.

This manuscript initially submitted to Nature has been reviewed by two experts. Both referees agreed that this work is interesting and novel, and deserves publication in certain form. However, both referees reached the same conclusion that the manuscript in the current status is not qualified for Nature due to the factors such as a relatively low T_c , and the lack of conclusive evidences for the topological superconductivity, etc. The referees have suggested the authors to resubmit the manuscript to either Nat. Comm. or PRL. By following the suggestion of the first referee, the authors resubmitted the revised manuscript to Nat. Comm. for publication.

I have read carefully the manuscript along with all correspondences between the referees and authors. I agree with the referees' assessments and judgments. In the revised manuscript and the corresponding reply letter, the authors have addressed properly the concerns and comments raised by both referees. Although the possibility of topological superconductivity remains largely speculative at the current stage, the experimental results presented in this manuscript are sufficiently novel and would attract immediate attention in the community of superconductivity. Besides, we should not expect to solve all issues in the first report of a new superconductor. I believe the manuscript in the current form should be able to fulfill the standard for publication in Nat. Comm.

Before this manuscript should be accepted, I hope the authors can address the following issues:

1. As pointed out by the authors, the thin film of Sr₃SnO on Si(001) in Ref. 18 exhibited a semiconducting behavior together with a weak ferromagnetism even at room temperature. Subsequent studies on the thin films (e.g. JAP 116, 164903 (2014)) have suggested that the transport properties are governed largely by the oxygen vacancies. The observed metallic normal state for the bulk material is in sharp contrast to that of the thin films. In addition, as mentioned in the manuscript, a 3% Sr excess or deficiency in the starting composition can reduce T_c to below 1.8 K. Since the revised manuscript focusing on the discovery of the first superconductor among the antiperovskite oxides is primarily material oriented, I believe it is mandatory to clarify the issues regarding the samples' chemical compositions. The authors should also provide more information, maybe in the supplementary materials, about how the physical properties vary in samples with different compositions or processed in different routes.
2. In the middle of Page 3, the authors have listed Ref. 10-14 as some examples of superconductors in antiperovskite carbides, nitrides, and phosphides. But it is

anti-post-perovskite rather than antiperovskite in Ref. 14. Besides, the page number of Ref. 14 is missing. The authors should either cited it as anti-post-perovskite or remove it.

3. First sentence of the abstract: either change "Investigation" to "Investigations" or change "have " to "has".
4. The second line of conclusion in Page 7, change "superconductivity" to "superconductor".
5. The second line under the subtitle "Topological superconductivity" in Page 10, change "elections" to "electrons".

REVIEWERS' COMMENTS:

Reviewer #4 (Remarks to the Author):

The authors have performed more experiments to study the influence of Sr stoichiometry on the physical properties of Sr_3SnO and added them in the Method section. Although it remains unclear about the exact content of Sr, the authors are sure that superconductivity occurs in the Sr-deficient samples. So, I recommend the authors change Sr_3SnO to $\text{Sr}_{3-\delta}\text{SnO}$ in the title and text. Other than that, I think the authors have addressed properly my previous concerns and the revised manuscript can be accepted for publication.

=====

Response to Reviewer #3:

We would like to thank Reviewer #3 for taking the time to read through our manuscript, and for providing us with comments to improve our work.

[Reviewer #3]

“The paper reports the synthesis of a possible new superconductor Sr₃SnO which has an anti-perovskite structure. The authors claim that it might be a superconductor with the Dirac metal feature in the normal state. A simple assessment on the data will find that the conclusions are not supported by the data and analysis. This does not allow me to give a positive judgement. Let me address the point below:”

[Our Response]

Reviewer #3 did not accept our claim to bulk superconductivity in Sr₃SnO, which is in contrast to the three other referees who judged the paper. He also raised the validity of the term “Dirac-metal” in describing Sr₃SnO, which we do not claim to prove in the manuscript, but is used to describe members of the antiperovskite oxides from band calculation evidence. These points are further addressed below:

[Reviewer #3]

“1. About the superconductivity. The XRD data seems to tell that the material Sr₃SnO might be successfully synthesized. The feeling from the XRD data would suggest a high percentage of the material of Sr₃SnO. While the DC ZFC magnetization would tell that the total superconducting magnetic screening volume at 2K is about 15%. In addition, a clear kink can be seen at about 3.7K which is most possibly due to the superconductivity of Sn impurity. The existence of Sn impurity has been confessed by the authors. Taking this diamagnetic signal away, one sees that the so-called diamagnetic signal from the possible superconductivity at 5K is really weak, which cannot be coherently explained as due to the major phase of Sr₃SnO.”

[Our Response]

The reviewer is concerned about the diamagnetic signal from Sn (with $T_c = 3.7$ K) dominating the total signal at 2 K. The presence of a small kink at 3.7 K is a valid concern that we had as well. However, as can be seen below from Fig. 2d [Previous Version] of the main text for batch C, the diamagnetic fraction at the point before the kink is already ~16%. Thus, the superconducting transition at 5.0 K is clearly distinct from the transition of Sn.

Furthermore, to resolve this concern, we have included recent data of a new sample with similar T_c and a higher volume fraction (batch D). This sample was synthesized in a slightly different way, which is now described in the METHODS section. For this sample, we observed a diamagnetic fraction of 62%, without any trace of Sn superconductivity (3.7 K). This data assures our claim of bulk superconductivity in Sr_3SnO . On the right, we show Fig. 2d of the revised manuscript.

We add a new author Jan Niklas Hausmann, who contributed to the synthesis of samples D and E.

[Reviewer #3]

“2. To prove the material is a Dirac metal, one needs elegant band structure calculation and spectroscopy experiment. Only the transport data cannot give a definite support to the conclusion.”

[Our Response]

In this paper, we do not claim to prove that Sr_3SnO is a Dirac-metal. However, the existence of Dirac-cone in the vicinity of the Fermi-level has been derived in the state-of-the-art band structure calculations by several groups, as well as by detailed analysis of orbital characters [Refs. 14, 15, and this work]. To further meet Reviewer #3’s request, we revised Supplementary Fig. 4: We newly added band structure results by HSE method and compare it with the standard band calculation (DFT-PBE), which was already shown in the previous version. Both support the existence of a Dirac-cone.

Spectroscopy experiments to clarify the band structure require single crystals with clean surface. We considered them as important future challenge.

We would like to thank Reviewer #3 again for providing us with comments to improve our work. We have sincerely answered to all your concerns, and made the necessary revisions including the results of new development in sample quality and of additional band calculations based on a more sophisticated method.

=====

Response to Reviewer #4:

We would like to thank Reviewer #4 for reading our manuscript as well as all the information provided, and providing encouraging comments. We have read through the Reviewer's comments carefully, and respond to them below.

[Reviewer #4]

“Dear Editor,

The manuscript reported the discovery of superconductivity with $T_c \approx 5$ K in an antiperovskite compound Sr_3SnO via measurements of resistivity and AC/DC magnetic susceptibility. Despite of the absence of obvious specific heat jump at T_c , the occurrence of bulk superconductivity has been substantiated by the observations of zero resistivity and the Meissner diamagnetism in this compound. This makes Sr_3SnO the first case of superconductor among the family of antiperovskite oxides. In addition, the authors have proposed a possible topological superconductivity based on the first-principles calculations and theoretical analysis.

This manuscript initially submitted to Nature has been reviewed by two experts. Both referees agreed that this work is interesting and novel, and deserves publication in certain form. However, both referees reached the same conclusion that the manuscript in the current status is not qualified for Nature due to the factors such as a relatively low T_c , and the lack of conclusive evidences for the topological superconductivity, etc. The referees have suggested the authors to resubmit the manuscript to either Nat. Comm. or PRL. By following the suggestion of the first referee, the authors resubmitted the revised manuscript to Nat. Comm. for publication.

I have read carefully the manuscript along with all correspondences between the referees and authors. I agree with the referees' assessments and judgments. In the revised manuscript and the corresponding reply letter, the authors have addressed properly the concerns and comments raised by both referees. Although the possibility of topological superconductivity remains largely speculative at the current stage, the experimental results presented in this manuscript are sufficiently novel and would attract immediate attention in the community of superconductivity. Besides, we should not expect to solve all issues in the first report of a new superconductor. I believe the manuscript in the current form should be able to fulfill the standard for publication in Nat. Comm.”

[Our Response]

Reviewer #4's response was positive towards publishing the paper in the current form. Below we respond to his comments and describe our revisions.

[Reviewer #4]

“1. As pointed out by the authors, the thin film of Sr₃SnO on Si(001) in Ref. 18 exhibited a semiconducting behavior together with a weak ferromagnetism even at room temperature. Subsequent studies on the thin films (e.g. JAP 116, 164903 (2014)) have suggested that the transport properties are governed largely by the oxygen vacancies. The observed metallic normal state for the bulk material is in sharp contrast to that of the thin films. In addition, as mentioned in the manuscript, a 3% Sr excess or deficiency in the starting composition can reduce T_c to below 1.8 K. Since the revised manuscript focusing on the discovery of the first superconductor among the antiperovskite oxides is primarily material oriented, I believe it is mandatory to clarify the issues regarding the samples’ chemical compositions. The authors should also provide more information, maybe in the supplementary materials, about how the physical properties vary in samples with different compositions or processed in different routes.”

[Our Response]

The contrast in the normal state transport data with the report on the Sr₃SnO thin-film is mainly due to Sr composition as eluded to in the previous manuscript. It should be noted, that the 3% excess or deficiency of Sr is in the starting composition, and does not correspond to 3% excess or deficiency from stoichiometric Sr₃SnO phase. All previously synthesized samples suffered from evaporation of Sr, which was difficult to accurately quantify. In some of the new samples we obtain results consistent with thin-film. In the METHODS we add the following description:

“An excess of 25% strontium was used and the glass tube was sealed under vacuum for synthesis of batch E. For this batch, a 100% yield of Sr₃SnO by weight with respect to the starting quantity of SnO was obtained. This batch E was considered an almost stoichiometric “Sr₃SnO” phase. Sample from batch E, with approximately stoichiometric composition, showed semiconducting behavior from transport measurement down to low temperature, with an anomaly at 4 K, suggesting an inclusion of a small superconducting region. However, in the magnetic measurement, this sample did not show evidence of superconductivity down to 0.15 K. The observed semiconducting behavior agrees with the result from Sr₃SnO thin-film reports (Ref. 17, 18), and supports our claim that the Fermi-level is shifted down in the bulk superconducting samples.”

In our revision, we added another author Jan Niklas Hausmann, who contributed to new data presented in the manuscript.

We add the subsequent thin-film work mentioned by Reviewer #4 as reference 18 in our revision. We are further investigating the effect of composition on the normal state properties and superconductivity, and these results will be published elsewhere when completed.

[Reviewer #4]

“2. In the middle of Page 3, the authors have listed Ref. 10-14 as some examples of superconductors in antiperovskite carbides, nitrides, and phosphides. But it is anti-post-perovskite rather than antiperovskite in Ref. 14. Besides, the page number of Ref. 14 is missing. The authors should either cited it as anti-post-

perovskite or remove it.”

[Our Response]

Following the recommendation of the Reviewer #4, Ref. 14 has been removed.

[Reviewer #4]

“3. First sentence of the abstract: either change “Investigation” to “Investigations” or change “have” to “has”.

4. The second line of conclusion in Page 7, change “superconductivity” to “superconductor”.

5. The second line under the subtitle “Topological superconductivity” in Page 10, change “elections” to “electrons”.”

[Our Response]

Corrections to the text were done according to these three comments. We thank Reviewer #4 for his/her careful examination of the manuscript.

We again thank Reviewer #4 for the positive assessment and thoughtful comments to improve our manuscript. In our revision, we have sincerely taken into account all your comments.

=====

The list of all changes made:

1. Page 1: A new author Jan Niklas Hausmann is added.
2. Line 1: A grammatical error is corrected.
3. Line 36: The irrelevant reference is deleted.
4. Line 61: Statement on the strontium composition is updated.
5. Line 72: Explanation of Fig. 2d is modified to include a new sample (Batch D).
6. Line 101: Statement on the comparison between Ref. 15's and our band-structure-calculation results is modified.
7. Line 111: Text is modified for accuracy.
8. Line 119: A grammatical error is corrected.
9. Line 123: Explanation of the synthesis procedure is itemized according to the samples.
10. Line 129: Explanation of the estimation of the strontium deficiency is added.
11. Line 131: Explanations of the synthesis procedure of the new samples (Batches D and E) are added.
12. Line 175: Text is simplified.
13. Line 181: A typing error is corrected.
14. Line 204: Additional acknowledgements are included.
15. Line 211: Author Contributions are updated to include JNH.
16. Page 17: The caption of Fig. 2 is updated to include the new sample (Batch D).
17. Page 5 of Supplementary Information: A new band-structure-calculation result is included in Fig. 4 (red curves), and the caption is modified accordingly.

Our response to the remarks by Reviewer #4:

We thank Reviewer #4 for reading our manuscript again. We have read through the Reviewer's comments and respond to them below.

[Reviewer #4]

The authors have performed more experiments to study the influence of Sr stoichiometry on the physical properties of Sr₃SnO and added them in the Method section. Although it remains unclear about the exact content of Sr, the authors are sure that superconductivity occurs in the Sr-deficient samples. So, I recommend the authors change Sr₃SnO to Sr_{3- δ} SnO in the title and text. Other than that, I think the authors have addressed properly my previous concerns and the revised manuscript can be accepted for publication.

[Our Response]

We changed Sr₃SnO to Sr_{3- x} SnO in the title and text. We kept Sr₃SnO when we are discussing the stoichiometric compound rather than superconducting one.

We again thank Reviewer #4 for the positive assessment and thoughtful comments to improve our manuscript. In our revision, we take into account all your comments.

During the revision, we came across a paper reporting superconductivity in La₃In with added Oxygen. The authors report that in La₃InO _{x} with a rather limited range of x beyond 0.3, the T_c is identical to that of La₃In (10 K) and strong and systematic reduction of the diamagnetic fraction occurs. Based on these results it cannot be claimed that antiperovskite La₃InO ($x=1$) is a bulk superconductor. In fact, this material is not included in the Superconducting Material Database (SuperCon) http://supercon.nims.go.jp/index_en.html . We included the information on this report in the introduction section of our paper and cited it as Ref. 10:
doi:10.1016/0925-8388(95)01909-X .